# Developing Village-Based Green Economy in an Endogenous Way: A Case Study from China

**DOI:** 10.3390/ijerph19137580

**Published:** 2022-06-21

**Authors:** Lili Li, Yiwu Zeng, Yanmei He, Qiuxia Qin, Jianhao Wang, Changluan Fu

**Affiliations:** 1School of Economics, Hangzhou Dianzi University, Hangzhou 310018, China; li42586@hdu.edu.cn; 2School of Economics, Hangzhou Normal University, Hangzhou 311121, China; 2018212212074@stu.hznu.edu.cn (Y.H.); changluanfu@hznu.edu.cn (C.F.); 3China Academy of Rural Development, Zhejiang University, Hangzhou 310058, China; 11922025@zju.edu.cn; 4China National Forestry-Grassland Economics and Development Research Center, Beijing 100714, China

**Keywords:** green economy, sustainable development, endogenous development, rural tourism, China

## Abstract

The idea of green economy is being taken seriously all over the world. For developing countries, the key to developing green economy is to strike a balance between environmental protection and economic development. As the largest developing country, China has been exploring scientific schemes to deal with the relationship between environmental protection and economic development. Developing rural tourism is an important way to transform ecological advantages into economic benefits. However, the role of rural tourism remains controversial. No scholars have yet provided solutions for village-level practices in developing countries from the perspective of endogenous development theory. Taking Yucun, a village in Zhejiang Province as an example, this paper reveals the endogenous way of green economy development at the village level through the method of case study. It is confirmed that the key to transforming rural ecological advantages into economic benefits is to cultivate the village’s endogenous development capacity, including activating local resources, cultivating local identity, stimulating local participation, and building a collaborative network. Only by implementing the endogenous development mode in rural areas cannot only stimulate the positive role of rural tourism and form a virtuous cycle, but also avoid the negative effects of rural tourism previously pointed out by scholars.

## 1. Introduction

At the beginning of last century, all countries were pursuing their own economic prosperity and revitalization. During this period, they completely ignored a series of negative impacts of economic development on the environment. Consuming resources and destroying the environment became a broad development strategy for all countries to become powerful at that time. It was not until some developed countries took the lead in modernization that they began to attach importance to environmental protection. As can be seen from the dynamics of the country rankings for sustainable development, the assessment of the dominance of developed countries since the 1990s assumes that environmental protection has become the first priority [1]. In contrast, despite remarkable economic growth over the same period, the deterioration of the ranking showed that China lacked effective environmental protection measures at that time. This is not unique to China, but a common feature of economic development in most developing countries [2,3]. Endless economic growth is impossible and should be stopped. Moreover, sustainable economic development and environmental protection are mutually reinforcing. Environmental protection is an important part of realizing sustainable economic development and one of the objective standards to measure the quality of sustainable economic development. The modern economy is increasingly inseparable from the support of environment and resources. Promoting sustainable economic development, protecting the ecological environment and effectively utilizing resources have become the consensus and common strategy of the green economy.

For developed countries, it seems easier to strike a balance between economic development and environmental protection. However, for developing countries, how to get rid of poverty while protecting the environment is a dilemma. Environmental protection and poverty alleviation are both important issues for governments in developing countries. The complex links between poverty and environmental degradation is still the focus of much research and debate [4]. However, there is a growing consensus, albeit in a rather piecemeal approach, on which policy instruments will be used to tackle both problems. The current thrust is mainly to pay for environmental services (PES) to correct market failures that lead to excessive environmental degradation [5,6,7]. On the other hand, conditional cash transfer (CCT) projects are being implemented to correct market failures that have led to underinvestment in social security in developing countries [8]. Although the above measures have been taken to alleviate the contradiction between economic development and environmental protection, they still cannot fundamentally solve the dilemma faced by developing countries. Existing policies are mostly based on the entire country’s environment and development issues, without in-depth analysis at the micro level. Most articles fail to find how villagers can play a greater role in achieving green economic development. To solve the dilemma between environmental protection and economic development, efforts should be made to extend the experience of one unit to the whole region.

As the world’s largest developing country, China is constantly exploring scientific solutions to deal with the relationship between environmental protection and economic development. On 15 August 2005, Comrade Xi Jinping, then Secretary of the Zhejiang Provincial Party Committee, inspected a village called Yucun in Anji County, Huzhou City, Zhejiang Province. Comrade Xi learned that Yucun had closed its mines and planned to turn to developing ecotourism. He highly affirmed this point and put forward a scientific thesis that “green mountains are gold mountains”. It can be said that Yucun is the epitome of China’s green development, which deeply proves the truth that “green mountains are gold mountains”. Developing rural tourism is an important way to transform ecological advantages into economic benefits. However, the role of rural tourism has always been controversial in academic circles. Some scholars pointed out the problems brought by the development of rural tourism, including the shock on traditional livelihood, destruction of local resources, excessive commercialization of culture, disappearance of agricultural knowledge, unfair distribution, investment loss, and so on [8,9,10,11,12,13,14]. The problems mentioned by scholars do occur frequently in some developing countries, and their doubts and concerns are justified. So, how can rural tourism in developing countries avoid these problems in the era of green economy? The theory of endogenous development provides important reference for this. However, no scholars have yet provided solutions for village-level practices in developing countries from the perspective of endogenous development theory. This paper tries to reveal the endogenous way of green economy development at the village level through a case study. It is confirmed that the key to transforming rural ecological advantages into economic benefits is to cultivate endogenous development capacity of villages, including activating local resources, cultivating local identity, stimulating local participation and building a collaborative network. Specifically, based on local resource endowment, Yucun does a good job in internal planning and external cooperation, cultivates villagers’ recognition of green economy and their role, and establishes effective villagers’ participation, so as to realize the rational utilization of local resources and create value-added. Only by implementing the endogenous development mode can the village avoid the negative effects of rural tourism previously pointed out by scholars and successfully achieve the transformation of ecological advantages into economic benefits. “Green mountains are gold mountains”, which is not only the way of butterfly transformation of Yucun, but also the successful promotion of China’s green economic development. The village-level practice of Yucun can also provide more concrete experience for developing countries on how to mobilize rural areas to achieve green economy.

## 2. Conceptual Background

### 2.1. Green Economy Concept and Its Spread in China

Green economy is a new economic development concept based on the concept of sustainable development, which is committed to improving human welfare and social equity, and greatly reducing environmental and ecological risks [15]. The term was first put forward by British environmental economist David Untidy. He proposed that green economy is an affordable economy built on social and ecological conditions [16]. Since then, the concept of “weak sustainability” with the theme of “economic growth can offset environmental and social losses” has rapidly formed a consensus in western developed countries. Western scholars have carried out a large number of studies on energy utilization efficiency and environmental pollution in economic development [17,18,19,20,21], and the concept of green economy has been extended in practice. The financial crisis around 2008 provided a historic opportunity for the rise of the green economy. The green economy has received unprecedented attention as the economic crisis has forced countries around the world to rethink economic policies and incorporate green and low-carbon concepts into fiscal plans to stimulate economic and job growth. Countries began to pay attention to the non-reduction of natural capital to enhance sustainability, which emphasizes the transition from passive governance to active investment and uses economic investment to promote the growth of ecological assets.

The concept of green economy has been widely spread around the world, and has been recognized and supported by more and more countries. The Chinese government not only recognizes the concept of green economy, but also attaches great importance to its implementation. However, the difficulty lies in promoting green development in rural areas. Farmers in developing countries often need a long process to accept the concept of green development due to their low overall quality. The primary problem facing the Chinese government is how to emancipate the minds of farmers and guide them to accept the new concept of green development. Therefore, to find a more appropriate way for rural residents to correctly treat the relationship between economic development and ecological environment is necessary. “Green mountains are gold mountains”, proposed by the Chinese government under President Xi, has brought the concept of green development to the countryside in an accessible way. On 24 August 2005, Comrade Xi published a commentary in Zhejiang Daily. It is mainly pointed that: green mountains can bring money, but money cannot buy green mountains; green mountains and gold mountains can produce contradictions and dialectical unity; when you cannot have your cake and eat it, you have to understand the opportunity cost and be good at making a choice; find the right direction and create conditions to turn green mountains into gold mountains; if ecological advantages are turned into economic advantages, such as ecological agriculture, ecological industry and eco-tourism, then green mountains are mountains of gold. The two-mountain slogan not only makes the concept of green economy deeply rooted in people’s hearts in a vivid metaphor and easily understandable way, but also emphasizes the responsibility of the governments of developing countries in promoting the green development.

### 2.2. Endogenous Development Theory and Rural Tourism Controversy

The endogenous development theory originated from the reflection and criticism of the external development mode in the 1960s and 1970s. Driven by economic globalization, profit-oriented capital has injected impetus into the industrialization and urbanization of various countries. The central position of cities has been continuously strengthened, while rural areas are increasingly facing problems, such as marginalization, hollowing out and involution. In this process, the exogenous development mode is widely used in the specific practice of rural development, and its core idea is that the intervention of external forces can stimulate rural development. However, its profit-seeking nature makes it become a force to plunder rural resources and aggravate rural depression. In the exogenous development mode, the intervention of external forces limits the autonomy of rural residents, ignores or even tramples on non-economic factors such as local humanities and ecology, resulting in the dilemma of subject loss and role alienation in the process of rural development [22]. In 1976, Japanese scholars put forward the endogenous development theory and believed that rural revitalization can be realized through endogenous development, that is, “people in different regions create it by self-discipline based on their inherent resources and heritage, drawing on foreign knowledge, technology and systems” [23]. Hereafter, UNESCO published a book named “Endogenous Development Strategy”, pointing out that endogenous development is generated internally and serves the people [24]. Different from the exogenous development, the endogenous development is essentially a self-directed development, which realizes the bottom-up transformation of rural development, highlights the local autonomy and initiative, gives full play to the role of the community, and emphasizes the decision-making power of local development choices, the control over the development process and the enjoyment of development benefits [25]. The internal realization path of endogenous development lies in: the first is to pay attention to local participation; the second is to cultivate local identity; the third is the protective development of local resources [26,27]. Ray believed that all development difficulties faced by rural areas could be improved by taking actions; however, rural development cannot completely rely on its own endogenous forces to achieve “pure” endogenous development, which was too idealistic in the context of globalization [28]. In addition to the premise of local participation, local identity, and local resources, the realization of the endogenous development of the village should also pay attention to the interaction and connection between the place and its environment [29]. In other words, the endogenous development mode should be a combination of both external and internal drivers.

Endogenous development theory provides useful reference for rural tourism. There has been controversy in academic circles about the impact of rural tourism development on rural community and farmers’ livelihood. On the one hand, many supporters believe that tourism can provide rural communities with a large number of employment opportunities and higher income, thus improving residents’ livelihoods and reducing their dependence on natural resources [30]. It can also change residents’ attitude towards biodiversity conservation and promote residents’ awareness and participation in environmental protection [31]. Ecotourism is regarded as an indispensable means of the conservation of natural resources and the development of local communities. The development of rural tourism is shaping a new relationship between man and nature, man and community [32]. It forces people to re-examine and update the traditional moral concepts and institutional arrangements [33]. On the other hand, opponents argue that ecotourism will lead to environmental degradation, destruction of wildlife habitat, economic loss, loss of income and unfair distribution, which will lead to the decline of residents’ willingness to protect [8,9,10,11,12,13,14]. The poor get little direct benefit from tourism, but bear a lot of costs [13]. The income from tourism often fails to offset the costs of tourism development. External forces of tourism development often have absolute control over local resources, so tourism will inevitably aggravate the stratification of residents in poor areas and widen the gap between the rich and the poor [14]. The rapid development of tourism will bring a series of new rural vulnerability problems, such as the deterioration of traditional social management structure in the community, the collapse of neighboring relations, the excessive commercialization of national culture, and the gradual disappearance of agricultural knowledge [10,12]. As a matter of fact, whether poor residents can benefit from tourism depends on what kind of tourism development mode they choose [34]. The endogenous development mode emphasizes the principles of local resources, local identity, and local participation. Therefore, only by implementing the endogenous development mode can the village not only stimulate the positive role of rural tourism and form a virtuous cycle, but also avoid the negative effects of rural tourism previously pointed out by scholars.

## 3. Research Methodology

This article adopts case study to reveal the endogenous way of green economy development at the village level. The case study method can be adopted when studying the formation of a phenomenon or the dynamic process of an event that involves various background conditions, multiple sources of evidence, and changing dynamics [35]. The case study method has unique advantages and values in focusing, displaying, and promoting specific practices [36]. Compared with quantitative methods, case study methods cannot only describe objective phenomena in detail, but also deeply analyze the reasons behind them, which is very helpful for researchers to grasp the essence of phenomena [37]. The object of this paper is the objective phenomenon that changes constantly in real life, which involves abundant conditions and multiple sources of evidence, and involves the question of “how” and “why”, so it is appropriate to adopt the method of case study.

The selection of cases should generally follow the principles of authenticity, representativeness, and feasibility [38]. Following these principles, this paper chooses Yucun, a village in China as an example. Yucun is regarded as the birthplace of “green mountains are gold mountains”. Yucun has experienced a process of destroying the environment, repairing the environment, and developing ecotourism to generate income. Yucun implements the endogenous development mode, so as to avoid the unreasonable phenomenon of rural tourism under the traditional mode. Yucun has successfully achieved a balance between environmental protection and economic development, enabling all residents to share the benefits of a green economy.

In order to ensure the accuracy of the information, we collected case data through a variety of ways, including in-depth interviews, participatory observation, and obtaining department reports. In this way, data from different sources form an evidence triangle, which can be compared and verified with each other, thereby ensuring the reliability of the research work. The field investigation was carried out in September 2020. We conducted in-depth interviews in a semi-structured way. The interviewees include 3 village cadres and 10 representatives of rural residents. Participatory observation refers to the in-depth observation, personal experience, and real-time consultation on the site of Yucun. Through careful field observation, data can be obtained and investigators can gain further insight into respondents’ working environments and behaviors [39]. In the process of field investigation, we also obtained the compilation of relevant reports and policy documents from the village committee of Yucun.

We performed data analysis concurrently during data collection to take full advantage of the flexibility of the case study methodology. We used our understanding of the endogenous development theory as a guideline to examine the initial data. Our guidelines and the related literature exposed new patterns and allowed us to further develop mappings for coded responses. As part of the data analysis, we performed open coding, axial coding and selective coding analysis on the raw data according to the programmatic grounded theory operating procedures proposed by Strauss and Corbin [40]. To increase the reliability of the research, the data analysis was carried out manually and independently by each researcher, who then compared notes. As for the conflicting data, we finally reached a consensus through group discussion and repeated screening.

## 4. Case Description

### 4.1. An Overview of Yucun

Yucun is located in Tianhuangping Town, Anji County, Huzhou City, Zhejiang Province, China (see Figure 1). Yucun is named after Yu Ridge, the remaining vein of Tianmu Mountain. Yucun stretches from east to west, surrounded by mountains on three sides. The Yucun Stream flows from west to east around the village, and the township road runs through the whole village. Yucun village covers an area of 4.86 square kilometers, with a mountain forest area of 4 square kilometers and a paddy field area of 0.39 square kilometers (all the numerical data about Yucun are provided by the village committee of Yucun). Yucun administers 2 natural villages, 1 central village, 8 villager groups, 280 households and a population of more than 1060. Since the opening of the mountain mining in 1976, the Yucun villagers have run their own cement plant relying on the high-quality limestone from the mountain. By the 1990s, the annual income of the village collective had reached more than 400 thousand dollars (the original monetary data in the text, Figure 2 and Figure 3 are converted by the authors using the average exchange rate of the year). However, the development of heavy industrialization has brought serious damage to the environment of Yucun. In 2005, Xi Jinping visited Yucun and put forward the saying that “Green mountains are gold mountains”, which made the villagers determined to develop green economy. Since then, Yucun has started a transition from selling stones to selling scenery. So far, Yucun has successively won the honors of National Democratic and Rule of Law Demonstration Village, National Beautiful and Livable Demonstration Village, National Ecological Culture Village, and National Civilized Villages. Yucun has become a model of beautiful village construction in China. Now, the vitality brought by environmental changes has been transformed into business opportunities in Yucun. Yucun has become a well-known “the paradise of health and wealth”. The village collective income of Yucun increased from 0.11 million dollars in 2005 to 1.04 million dollars in 2020, and the per capita disposable income of Yucun increased from 1065.96 dollars in 2005 to 8071.32 dollars in 2020 (see Figure 2 and Figure 3).

### 4.2. The Development Process of Yucun

According to the relationship between environmental protection and economic development, the development process of Yucun can be divided into three stages. In the first stage, Yucun gained economic benefits at the expense of sacrificing the ecology. In the second stage, the villagers determined to restore the ecology and lay the foundation for the industrial transformation. In the third stage, Yucun successfully transformed its ecological advantages into economic benefits.

#### 4.2.1. The First Stage: Relying on the Mountain and Abusing Resources

Since the 1980s, villagers have built more than a dozen resource-based economic entities, such as lime factories, brick factories and cement factories, using high-quality limestone resources. Relying on the stone economy in mountainous areas, more than half of the villagers have solved the employment problem. By the 1990s, the monthly income of village’s workers had reached more 1 thousand dollars per capita, and the annual income of the village collective remained around 313.8 thousand dollars. Yuncun became the richest village in Anji County. However, the ensuing environmental pollution, ecological damage, and frequent safety accidents made people taste the bitter fruit of predatory development (see Figure 4; the pictures in Figure 4 and Figure 5 are provided by the village committee of Yucun). Several limestone mines blast mountains more than 100 times almost every day. The mountains were devastated. The sky was covered with smoke and dust for many years. The sand and stones were flying in the bamboo forest. The river was flowing with white mud. The stream water was filthy. Even the Millennium ginkgo trees did not bear fruit. “When I went to work, I couldn’t see the blue sky”, “When I came back, my eyebrows and hair were white and covered with dust”, “I didn’t dare to open the window at home and had no place to hang clothes. On rainy days, the village streams were like soy sauce and the village roads were muddy and potholes”, the villagers recalled. The successive mine casualties have hurt the people of Yucun. Many villagers also suffered from respiratory diseases called “stone lung”.

#### 4.2.2. The Second Stage: Shutting down the Factories and Restoring the Mountain

People in Yucun deeply feel that developing the economy at the expense of the environment is not a long-term solution. High-polluting enterprises such as mines and cement plants must be strictly shut down. However, it is very difficult to implement. The ideological transformation of cadres and villagers is facing tremendous difficulties, resistance, and pressure. The mine had been closed for five or six years under the villagers’ hesitation. In March 2005, under the leadership of Bao Xinmin, the secretary of the village party branch, Yucun closed some mines and factories to explore green development. At the beginning of the closure of the mine, the collective economy and villagers’ income of Yucun declined significantly. Before the coal mine and cement plant closed, more than half of Yucun’s 280 families worked in the mining area. In 1992, a miner in Yucun could earn about 150 dollars a month. In 2004, the collective income of Yucun was 0.38 million dollars, but just one year later, this figure plummeted to 0.11 million dollars. The pain of transition made some people hesitate.

On 15 August 2005, Xi Jinping put forward a famous slogan in Yucun that “green mountains are gold mountains”. Since then, Yucun has completely shut down all mines, cement factories, and a large number of bamboo chopsticks enterprises, cutting off environmental pollution sources, and leaving time and space for ecological self-repair and recuperation. Pumice stones on main slopes and other slope sections should be cleaned to eliminate potential safety hazards such as slope falling caused by dangerous rocks and pumice, so as to create good conditions for greening of slopes. Through ecological restoration, the exposed rocks presented the beauty of green, which complemented the surrounding ecological environment, and effectively improved the ecological environment of the mining area. Under the strong support and careful guidance of the transportation department in Anji, Yucun has widened, hardened, and upgraded the township road from the original 4 m wide to 12 m wide. The widening adds sidewalks to both sides, separating the roads, fields, and houses. The villagers also made overall planning and design for the houses along the road, and the scenery along the rural roads was reproduced and integrated with the road scenery.

Ecological restoration usually takes several years. The development of tourism from input to output also needs a process. In order to solve the employment problem of the villagers during this transitional period and ensure that the income level of the villagers did not decrease, more than 50 family workshops and factories for bamboo processing were established in Yucun. This is an emerging industry that will not cause pollution to the ecological environment. In addition, these workshops and factories can provide tourists with local products and have become part of the tourism development of Yucun.

#### 4.2.3. The Third Stage: Turning Ecological Advantages into Economic Benefits

Yucun has scientifically laid out ecological, production, and living space, becoming one of the first pilot villages of rural planning and construction in Zhejiang Province. Yucun accelerated the reconstruction of the old village, and took the lead in building 7 residential villa model houses with the village advance payment. A new planned community was built to solve the resettlement of some rural households. Taking the opportunity of the beautiful village construction, ecological civilization construction and the world bank loan project, Yucun has carried out six major actions in an orderly manner, including factory renovation, road upgrading, river improvement, sewage treatment, waste classification, and agricultural rehabilitation. On this basis, Yucun guided and encouraged villagers to develop agritainment, and completed the construction of an agritainment service center. The village collective has invested more than 0.63 million dollars to develop the tourist attractions in Longqing park. Through villagers’ autonomy and market participation, Yucun has made every effort to improve the quality of scenic spots, rafting, reservoirs and parks, and has gradually formed a leisure tourism industry chain of river rafting, outdoor development, leisure meetings, mountaineering and fishing, fruit and vegetable picking, and agricultural experience. From absorbing mountain resources to enrich the mountain, the appearance of Yucun is changing with each passing day. The Mine Relic Park was built on the formerly bombed cold water cave mine, the two-mountain greenways were built on the uneven roads of the village, as well as eco-tourism projects, such as Lotus Mountain rafting, pastoral picking farms, and so on. The two-mountain scenic spot in Yucun has become a national 4A scenic spot with ecotourism, beautiful livable area, and pastoral sightseeing area. It is China’s first eco-tourism and rural resort themed around the practices of two mountains (see Figure 5). The concept of ecological civilization has been written into village regulations, folk conventions, and family precepts. The environmental advantages of green mountains have actually been transformed into the practical productivity of the construction of gold mountains. The number of agritainment in Yucun increased from 10 in 2015 to 42 in 2019. In addition, the number of tourists visiting Yucun increased from about 100,000 in 2015 to about 900,000 in 2019 and 2020 (see Figure 6). Many positive changes have taken place in Yucun (see Table 1). On 30 March 2020, General Secretary Xi visited Yucun once again after 15 years. Xi said that, “As time flies, I can still vividly remember the circumstances of that year. Now, Yucun looks completely different. The dream of beautiful village construction has become a reality in Yucun. The practice of Yucun have proved that green development is absolutely correct. If you choose the right path, you must stick to it”.

## 5. Case Findings

### 5.1. The Results of Data Coding

#### 5.1.1. Open Coding

Open coding is to conceptualize and categorize the collected data for identifying phenomena, defining concepts and discovering categories. Conceptualization is the breaking down of source material and naming it after separate stories or events, while categorization is the process of classifying concepts related to the same phenomenon. This step is to use concepts and categories to correctly reflect the content of the original data, and in the process of continuous comparison, the abstract concepts and categories are constantly revised. Centering on the research theme, we try to suspend the influence of personal and academic prejudice as much as possible. We disassemble, refine, and give concepts to the original data, and then recombine them with preset logic. A total of 46 concepts are formed (see Table 2). On this basis, we do further induction and abstraction, forming 13 categories (see Table 2).

#### 5.1.2. Axial Coding

Axial coding is to discover the potential logical connections between categories and develop the main category and its subcategories. Based on 106 concepts and 13 categories obtained by open coding, we conducted detailed comparison and analysis of each category repeatedly, so as to dig into the relationship between categories and summarize scattered categories to establish categories. Through the above data analysis process, we identified 4 primary categories based on the above 13 categories, which are: Activating local resources (AA1), Cultivating local identity (AA2), Stimulating local participation (AA3), and Building collaborative network (AA4) (see Table 3).

#### 5.1.3. Selective Coding

Selective coding is to mine the core category from the main category, analyze the connection between the core category and the main category and other categories, and describe the behavior phenomena and context conditions in the form of a story line. After the plot is completed, a new theoretical framework will be developed. Through the data analysis, it is found that the key to the success of green development in Yucun is to adhere to the path of endogenous development. Endogenous development should be the core category of green economic development at the village level. The four main categories are parallel, and they all belong to the sub-dimensions of the core category. In other words, for villages in developing countries to achieve effective green development, measures from all four aspects, including activating local resources, cultivating local identity, stimulating local participation, and building a collaborative network, must be taken simultaneously. We summarize 13 categories obtained from open coding, 4 main categories formed by axial coding and 1 core category formed by selective coding into a whole, namely the endogenous development framework of Yucun’s green economy, as shown in Figure 7.

#### 5.1.4. Saturation Test

Theoretical saturation means that there are no new concepts, categories or relations in the new data, and the concepts or categories abstracted from the research are enough to cover the data collected by researchers or even the new data. In order to test whether the concepts and categories refined in this study have reached theoretical saturation, a random extraction method is adopted to verify the theoretical saturation. That is, two thirds of the data are randomly selected from the data pool for coding analysis and the remaining one third for the theoretical saturation test. It is found that the categories in the model have been developed very rich, and no new concepts and categories have emerged, indicating that the research has reached theoretical saturation.

### 5.2. Further Interpretation of the Findings

#### 5.2.1. Activating Local Resources

The endogenous development theory believes that the first step to development is to utilize the local resources. The endowment of resources often determines the development path and direction of a region. The realization of rural endogenous development depends on the integrated utilization of internal and external resources. Generally speaking, internal resources are provided by local residents, and external resources are provided by governments and non-governmental organizations. The resources provided by locality include natural resources and human-centered resources, such as culture, social relevance, humanistic quality, and other humanistic resources [41]. In the process of promoting local development, rational and effective use of local ecological and cultural resources is an important prerequisite for realizing green economy. In addition, villagers’ entrepreneurship is an important factor to activate local resources. REST (recreation, entertainment, sport, and tourism) is an entrepreneurial process where innovation and change are key factors in seizing opportunities, gaining competitive advantage, and developing rural tourism. REST can be an important factor in endogenous development, especially in rural areas with a low degree of urbanization.

Yucun is rich in landscape resources. Not only is it surrounded by mountains on three sides, but its reservoirs and streams are clear (see Figure 8). Its advantages in natural resources are more obvious after abandoning mining economy and restoring mountains and forests. It has superior geographical location and great tourism potential. Yucun is located in a traffic circle of two hours’ drive from Shanghai, Suzhou and one hour’s drive from Hangzhou (Shanghai, Suzhou and Hangzhou are the major cities in China’s Yangtze River Delta region. The Yangtze River Delta region is one of the largest urban agglomerations in China, where a large number of potential customers for rural tourism live). Relying on the advantages of beautiful ecological environment and geographical location, integrating agricultural historical, and cultural resources has become the main direction for Yucun to promote green development. In accordance with the standardization requirements for the construction of beautiful villages and national scenic spots, Yucun has continuously integrated the green economy concept into all fields of village construction, optimized and adjusted the village development plan, carried out the construction work in an all-round way, and gradually formed the village spatial layout of “three districts and one circle”. The ecotourism area takes mountain hiking and Lotus Mountain scenery as the core, and takes leisure tourism as the main project. The beautiful livable area is centered on agritainment, with a comfortable living environment and more than 90% of the green courtyards in the village. The pastoral sightseeing area is centered on picking fruits and vegetables and appreciating the sea of flowers, providing villagers and tourists with natural beauty and pastoral fun. The green way surrounds the village and connects the various scenic spots of Yucun. These high-quality environmental resources have been successfully transformed into tourism resources. The improvement of the environment of Yucun attracts more and more young people to return home to start a business in REST. For example, Hu, a young entrepreneur who has returned to Yucun, now has a new identity as “Yucun Partner”. With his expertise in design, he has set up a design studio to help the villagers improve their agritainment and boost the development of the tourism industry in Yucun.

According to the endogenous development theory, in the process of local endogenous development, it is necessary to pay attention to the role of super-local factors [41]. In the embryonic stage of local development, super-local factors can become a catalyst for the development of local resources. It is very necessary to invest the capital, technology, and information. For the rural areas marginalized in the era of industrialization and informatization, the development of local resources needs to be stimulated and supported by the governments and other social forces. Making full use of the opportunity of the construction project of Zhejiang’s beautiful countryside, Yucun has improved the village’s roads, networks, and other infrastructure facilities, laying a solid foundation for the development and utilization of tourism resources. According to their own needs, village collectives have carried out in-depth research on the central and local policies supporting and benefiting agriculture. Especially under the background of Zhejiang Province’s implementation of the Thousand Villages Demonstration Project, the Ten Thousand Village Renovation Project and the Five Rivers Co-governance Project, Yucun vigorously renovated the appearance of the village, took the lead in formulating village rules and regulations, and formed a systematic garbage classification and sewage treatment practice, which brought the appearance of the village closer to the standard of the scenic spot. Over the past 10 years, Yucun has invested more than 8 million dollars in infrastructure construction and environmental improvement. A series of powerful and effective measures have brought profound changes in the environment of Yucun.

#### 5.2.2. Cultivating Local Identity

In order to better solve the intricate relationship between local forces and super-local forces, the concept of “territorial-cultural identity” was constructed based on local culture, history, and material materials [42]. In rural development, local residents are the main forces. It is very necessary to activate their subjectivity in consciousness through territorial-cultural identity. The territorial-cultural identity helps local residents participate in decision-making and supervision of the development path, thereby further promoting the formation of local development autonomy [29]. The territorial-cultural identity also provides common emotions and goals for local residents, and strengthens the bonds needed to integrate various local forces in addition to economic interests. Territorial-cultural identity endows this development with cultural significance that cannot be brought about by the increase of material wealth alone, and is an important basis for sustainable development. The sources of identity construction include not only rural cultural materials and landscape, but also the authority of collective leaders and the sharing of development interests.

The establishment of local identity requires village cadres and superior organizations to guide villagers to establish correct value judgment and identity to Yucun. The two-mountain slogan makes the concept of green economy deeply rooted in people with vivid metaphor and easy to understand. However, the implementation of green economic development cannot only rely on external forces to promote to achieve. In many cases, it is necessary for villagers to establish a sense of local identity. In the process of establishing local identity, the village party branch played a good leadership role. The village party secretaries and other village cadres take the lead in guiding villagers to close mines, protect the ecological environment, and promote green economic development. At the same time, Yucun has deeply practiced the concept of green economy, practically changed its development ideas, promoted the transformation of village scenic spots and the transformation of resources into shares. The villagers can earn wages, rents, shares, and profits. With the continuous increase of villagers’ income and optimization of the environment, the villagers’ recognition of green economy is also growing. Because every household in the village has brought up the “green rice bowl”, the people in Yucun are more aware of the importance of a clean and tidy village environment. In July 2017, Yucun issued a new “Yucun Regulations and People’s Agreement”, which supplemented and refined the content of environmental protection. Yucun specially built the old mining area into a mine relic park, allowing the people of Yucun to perceive the significance of the green economy concept from the historical changes. Without the understanding of the villagers, there would be no measures to shut down mines and factories. Without the efforts of the villagers, there would be no gorgeous transformation of Yucun today.

#### 5.2.3. Stimulating Local Participation

Rural endogenous development is a process of participation. Through participation, local residents express their interests and appeals, and have an effective impact on the decision-making [24]. Moreover, local residents also transform themselves into the core force to promote endogenous development through participation [25]. Endogenous development theory holds that two basic conditions must be established in order to ensure the influence of residents’ suggestions on decision-making: one is to ensure an institutional environment for local residents to fully express and comment; the other is a fixed activity place required by local residents [43]. In short, this participation process should be an organized collective behavior under the framework of institutional guarantee, and its opinion statement process is highly autonomous and public. Romina believed that through organizing forums, residents’ united organizations provided an important opportunity and occasion for local subjects in different fields to express and comment, and effectively promoted relevant stakeholders to participate in the decision-making process of local development [43].

In the process of developing green economy, Yucun pays much attention to the livelihood dimension of rural construction, cultivates ecological consciousness, and action participation of every villager. Yucun implements the system of “four democracies and three openness”, namely, democratic election, democratic decision-making, democratic management and democratic supervision, and openness of party affairs, village affairs, and finance. In addition, the village collective pays attention to observing public opinion, guiding the masses to participate in the construction of beautiful villages, and mobilizing the participation enthusiasm of the masses to the greatest extent, so as to create a great atmosphere of beautiful countryside for the whole people. Yucun has set up a villager council composed of retired cadres, a joy and bereavement events council composed of villagers’ group leaders, a moral evaluation council composed of villagers’ representatives, an anti-drug and gambling council composed of women representatives, and a peace volunteer team composed of some party members and some villagers, so as to expand channels for villagers to play their roles, and form the rural governance mode based on rule of law, virtue, and autonomy. Regarding major issues, the village collective convened a villagers’ congress for democratic consultation and joint decision-making to ensure that the decision-making and implementation not only comply with laws and regulations, but also reflect the overall will of the villagers. Yucun has fully mobilized the consciousness and enthusiasm of villagers’ participation and promoted the harmonious development of the village. Since 2005, there has not been a single criminal case, not a single group visit, not a single leap-level petition, not a single village official has been punished by law. The practice of Yucun manifests the spirit of relying on the masses. While continuously satisfying the villagers’ growing needs for a beautiful environment and a better life, it has mobilized the villagers’ enthusiasm, initiative, and creativity to voluntarily participate in the development of green economy. The sense of belonging and responsibility of hometown is a strong support for villagers to participate in environmental governance. Over the years, Yucun has organized a number of teams including party members and volunteers to promote sewage treatment and ecological river construction and made remarkable achievements. In addition, Yucun continues to improve the rural social security system, popularize rural cooperative medical care, and endowment insurance, and increase care for needy families and vulnerable groups in the village. All in all, Yucun tries to make the achievement benefits for every villager as equally as possible, so as to stimulate the participation enthusiasm of all villagers.

#### 5.2.4. Building a Collaborative Network

Endogenous development theory holds that rural development should neither rely excessively on external forces nor overly self-enclosed, but should be a process of comprehensive action of internal and external forces in rural areas. Therefore, attention should be paid to the interaction between internal and external subjects, and the integration between internal and external resources. Therefore, it is quite important to establish a network of collaborative governance. Collaborative governance, as an emerging governance method, advocates bringing multiple stakeholders together to form a consensus-oriented method of participation in decision-making to solve complex environmental governance issues, thereby forming a complete collaborative network. Collaborative governance refers to a governance mode in which multiple stakeholders make decisions together on a common issue and reach an agreement by sharing rules, guidelines, and frameworks [44]. Collaborative governance has the following characteristics: problem-solving-oriented and focusing on solving regulatory problems; stakeholders involved in all stages of the decision-making process; solutions and rules are temporary and can be revised; accountability goes beyond traditional public and private roles in governance; establishing a flexible and open agency to act as the convener and promoter of multi-stakeholder negotiations [45].

Yucun has made good use of collaborative governance in the development process of green economy. Yucun itself has good initial conditions for collaborative governance. In the transformation process, the government plays a supportive role in the collaborative governance of rural human settlements. Regarding the issue of rural living environment, the government has adhered to the chief responsibility system and implemented the promotion mechanism at various levels. The government has continuously strengthened preferential policies and financial support. According to statistics, over the past ten years, Zhejiang provincial governments at all levels have invested more than 30 billion dollars in village renovation and beautiful rural construction, which has effectively mobilized the enthusiasm and initiative of grass-roots governments. For example, garbage is the main problem in the rural living environment. Zhejiang was the first province in China to implement the credit system for the classification of rural garbage. It has successively issued the 13th Five-year Plan for the Construction of Harmless Treatment Facilities for Domestic Garbage in Zhejiang Province, the Five-year Action Plan for the Classification of Domestic Garbage in Zhejiang Province, and the Measures for the Management of Food Garbage in Zhejiang Province, which would earnestly solve the problems that are most closely related to the people.

Yucun also fully stimulated the vitality of market players and implemented effective integration. By stimulating market players, Yucun concentrated a variety of human and material resources into the operation of rural residential environment improvement project, thereby strengthening the effective integration of agricultural resources. Yucun has a good business environment. On the one hand, Yucun introduces a series of measures to support entrepreneurship and innovation, actively supporting the growth plan of high-tech enterprises. On the other hand, Yucun keeps on innovating and transforming the mode to fully mobilize the enthusiasm of market players. In the process of introducing multi-party cooperation, a complete benefit distribution mechanism is established to realize the multi-party benefits of village collectives, operating companies, market capital, and villagers. With the continuous investment of market capital, the industrialization and capitalization of green economic development in Yucun develop rapidly. An emerging trend is that rural tourism in Yucun will evolve into green economy clusters and promote the development of small and medium-sized enterprises. More and more investors, cooperative enterprises and returning youth come to Yucun for investment and entrepreneurship. The green industrial chain of Yucun is gradually improved. There are producers of green agricultural products in the upstream, processors, and designers of green products in the middle, and sellers and transporters of green products in the downstream. The cooperation network based on a cluster makes the green economy of Yucun have more obvious advantages and more sustainable.

More importantly, the villagers in Yucun have played the main role in collaborative governance. Yucun respects the wishes of the villagers and the actual needs for the living environment, and constantly improves the guidance mechanism for villagers’ participation. As for the project of unified planning, villagers can participate in the characteristic landscape and agricultural industrial park of the village as shareholders, which improves the village’ s living environment, and, at the same time, enhances the endogenous motivation and the consciousness of villagers’ participation in collective economic construction. For example, the local government implemented a garbage-for-goods policy in Yucun and established a garbage exchange station in the village. The villagers picked up cigarette butts or bottles on the road in exchange for eggs and other daily necessities. The villagers change the village environment bit by bit with their hands. Seeing the environment in the village getting better day by day, they become more conscious as the guardian of the environment. In addition, the cooperation between the government and the villagers has been continuously strengthened. Through these targeted village governance schemes, the government cannot only enable the villagers to quickly reach the cooperative intention in the process of collaborative governance of rural residential settlements, but also effectively stimulate the enthusiasm, initiative, and creativity of villagers’ participation.

## 6. Discussion

Most countries in the world have gone through an extensive development path that destroys the environment and consumes resources, resulting in many negative problems. With the continuous development of economy, the government began to attach importance to environmental protection. However, compared with developed countries, developing countries are still in a dilemma in terms of economic development and environmental protection. Developed countries have a higher level of national income, which makes it easy to achieve a win-win situation of ecology and economy. How to increase economic income while protecting the ecological environment is a problem that many developing countries need to solve. Most of the measures currently taken by developing countries are from a macro perspective, lacking of exploration on the rural level. Rural residents in developing countries are generally less aware of environmental protection. Their awareness of green economy is still very weak, and their willingness to develop green economy is not strong. This is one of the major constraints for developing countries to promote green economic development in rural areas. As the largest developing country in the world, China has made positive progress in transforming its ecological advantages into economic benefits. The two-mountain slogan put forward by General Secretary Xi Jinping is a high generalization of the combination of ecological environment and economic development, which makes that the idea of green economy quickly received the attention of local governments and was easily accepted by rural residents. However, it is only a starting point. This is not a sufficient condition for successful village-level practices. As the birthplace of the two-mountain slogan, Yucun unswervingly explored and practiced the correctness of this development path, and finally proved its feasibility to all villages. As a model of China’s implementation of ecological economization, Yucun expounds the harmonious and unified relationship between economic development and environmental protection. Relying on its own resource advantages, Yucun made a breakthrough in rural construction under the premise of scientifically formulating development strategies. Meanwhile, Yucun gave full play to the positive leadership role of village cadres, guided the masses to actively participate in the construction, and paid attention to the development of rural industries. It has cooperated with the government and the market in many aspects to form a cooperation network, which has promoted the green development of Yucun.

There has been controversy in academic circles about the impact of rural tourism development on rural community and farmers’ livelihood. Yucun’s practice provides a new perspective for these disputes. That is, whether rural tourism can play a positive role depends on the choice of the development mode. The endogenous development theory emphasizes that development paths should be chosen by local people, development process should be controlled by local people, and development benefits should be enjoyed by local people [28]. Only by implementing the endogenous development mode can the village not only stimulate the positive role of rural tourism and form a virtuous cycle, but also avoid the negative effects of rural tourism previously pointed out by scholars. Through the case study of Yucun, we identify the endogenous development mechanism of green economic development at the village level, including activating local resources, cultivating local identity, stimulating local participation, and building a collaborative network. The importance of these endogenous development mechanisms and the representative views of previous scholars on the positive and negative effects of rural tourism are summarized in more detail in Table 4. In general, the case of Yucun provides some inspiration for developing countries to promote a rural green economy from the perspective of endogenous development. The implementation of green economy is inseparable from the provision of sufficient external resources and super local forces by the state for rural development. However, in this process, villagers must fully understand the advantages and disadvantages of local resources and make sustainable use of local resources to ensure that they are not overexploited. The effective participation of local residents is crucial to the implementation of green economic development. Although the joint participation of external forces can make up for the limitations of local forces, the conflict between the two forces needs to be communicated and negotiated through an institutionalized framework under the premise of taking local residents as the leader. In this process, local residents need to have a clear voice, and their rights to make suggestions should be fully respected. Stimulating multi participants’ recognition of rural development not only provides a psychological link for the integration of multi forces and promotes their participation, but also transforms the process of rural development into a sustainable value creation process.

The theory of endogenous development was put forward in the second half of the 20th century when Europe was facing the problems of rural marginalization, depletion, and hollowing brought by globalization and free capitalism, and was widely used to guide the practice of European rural development. The research of this paper shows that the endogenous development theory is also applicable to the practice of green economic development at the village level. On the basis of recognizing the three basic elements of resources, participation and identification as the core elements, the endogenous development theory takes internal and external resources and the connection between local and super-local relations as the starting point and emphasizes that the decision-making power of local development options, the control of development process, and the right to enjoy development interests are the keys to ensure rural development. The research findings of this paper are generally consistent with the theoretical framework, which undoubtedly strengthens the explanatory power and comprehensiveness of endogenous development theory.

It must be emphasized that it is not easy for a village to adhere to the endogenous development path in the process of developing green economy, because it is a systematic project. Assessment and utilization of local resources is the premise of endogenous development of green economy in villages. However, the local resources must be multi-dimensional rich resources, involving natural, human, historical, geographical, policy and various aspects of resources. If a village has a single resource, then developing ecotourism is likely to fail and eventually suffer losses. To enhance the villagers’ recognition of developing green economy, it also needs the villagers’ empowerment in various aspects, including economic empowerment, psychological empowerment, and social empowerment. The economic benefits positively influence villagers’ confidence in the development of tourism in their communities and their attitudes towards the environment [50]. Gupta and Bovarnick argued that local people will have an incentive to conserve natural resources only if they reap a significant share of these benefits [51]. Ecotourism can empower local people psychologically if they observe and respect traditional cultures and customs [52]. Social empowerment refers to strengthening the cohesion and integrity of communities through activities such as ecotourism [53]. Participation is another core element of rural ecotourism. Numerous studies have shown the importance of seeking out the views, values, and interests of community residents when planning a viable tourism strategy for an area [54]. More and more scholars believe that the real link between ecotourism and environmental protection comes from ownership and management participation, not just through economic interests [50]. If local people are empowered and motivated to participate in the decision-making process, they will be more willing to work to protect the local environment and develop ecotourism [50]. The participation of villagers helps build confidence in long-term sustainability and ensure the sustainable use and management of natural resources, reducing the adverse impact of man-made activities on the environment. Unlike traditional agriculture, tourism requires high skills and market knowledge to provide satisfactory tourism services to tourists. However, these knowledge and abilities are often not previously possessed by local residents, and the lack of education and skills is an important obstacle for residents to participate in tourism. At this point, cooperation with external agents becomes necessary. It should be noted that although the village ecotourism development needs the intervention of external forces, three misunderstandings should also be avoided. First, the wrong external forces cannot effectively activate the local ecotourism resources and cannot bring substantial development to the local area. Second, it relies too much on external forces and lacks the cultivation of local forces, resulting in insufficient self-evolution ability and unable to achieve sustainable development. Third, commercial capital groups take government subsidies and incentives as the fulcrum to plunder local resources and interests. In a word, external subjects mainly play the role and responsibility of service providers and promoters.

## 7. Conclusions

How to correctly handle the relationship between economic development and environmental protection has always been a problem that many countries try to solve. Especially for developing countries, there are many difficulties in achieving the synchronization of economic development with ecological civilization. In recent years, some governments of developing countries have taken some macro measures to adjust the relationship, but the effect is limited. How to balance environmental protection and economic development, especially at the village level? A case study from China gives the answers. As the birthplace of the two-mountain slogan, Yucun makes use of its own resources and makes a breakthrough in rural construction according to local conditions under the premise of scientifically formulating a development strategy. Taking Yucun of Zhejiang Province as an example, this paper reveals the micro mechanism of developing green economy at the village level. It is confirmed that the key for the transformation of rural ecological advantages into economic benefits is to cultivate the village’s endogenous development capability, including activating local resources, cultivating local identity, stimulating local participation and building a collaborative network. The conclusion of this paper not only provides empirical evidence for turning ecological advantages into economic benefits in rural developing countries, but also provides a new theoretical perspective to examine the role of rural tourism. The practice of green economic development should rely on different resource endowments in various regions to promote the formation of industries with local characteristics. The government should guide local villagers to give full play to their own advantages. All the villagers should actively participate in the construction of rural residential environment, continuously broaden industrial development channels, and extend the industrial chain through multi-party linkage of collaborative governance subjects, so as to make resource elements continue to produce economic benefits. The practice of Yucun presents the significant spontaneous and endogenous characteristics, which means fully respecting the dominant position of the villagers and giving full play to the creativity of the grass-roots masses. Therefore, in the process of practicing green economic development, only by making good use of local resource advantages, mobilizing collective active participation and establishing everyone’s sense of identity, can the green economy be applied to different regions more effectively, better solve the problems that have plagued everyone for a long time, and realize the ecological economization.

## Figures and Tables

**Figure 1 ijerph-19-07580-f001:**
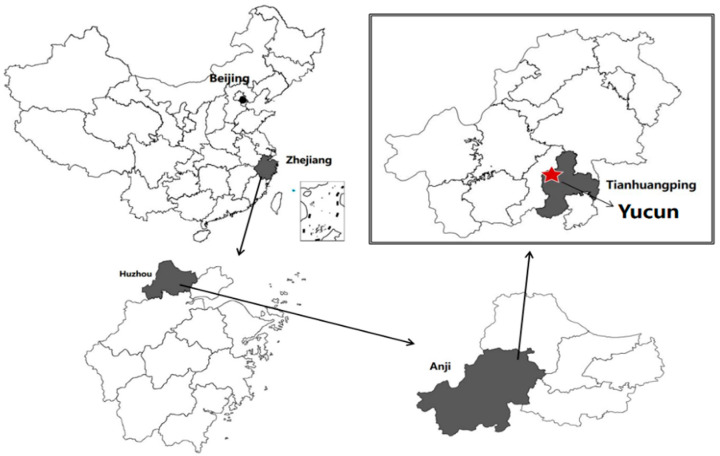
Geographical location of Yucun.

**Figure 2 ijerph-19-07580-f002:**
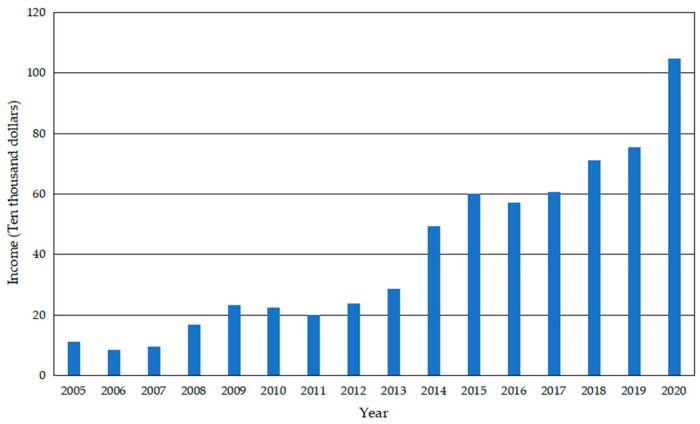
Village collective income of Yucun from 2005 to 2020.

**Figure 3 ijerph-19-07580-f003:**
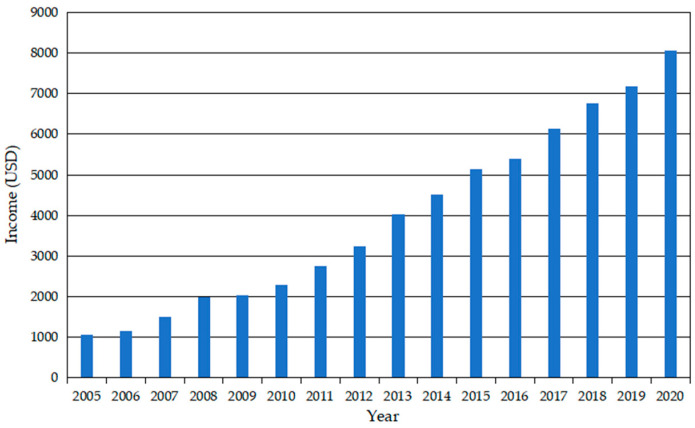
Per capita disposable income of Yucun from 2005 to 2020.

**Figure 4 ijerph-19-07580-f004:**
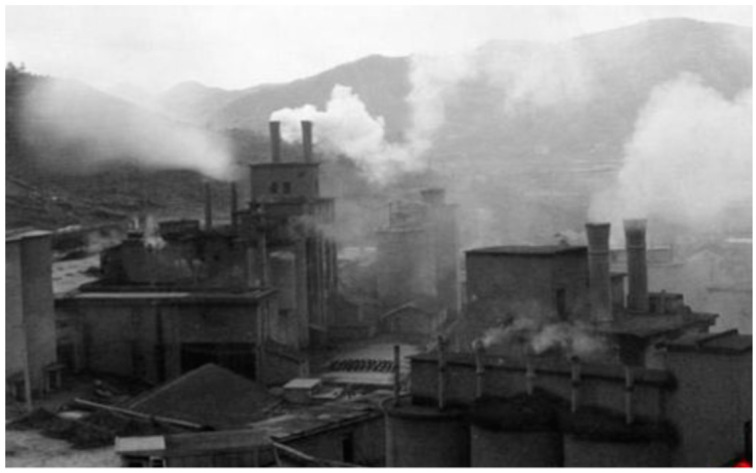
Appearance of Yucun in the 1990s.

**Figure 5 ijerph-19-07580-f005:**
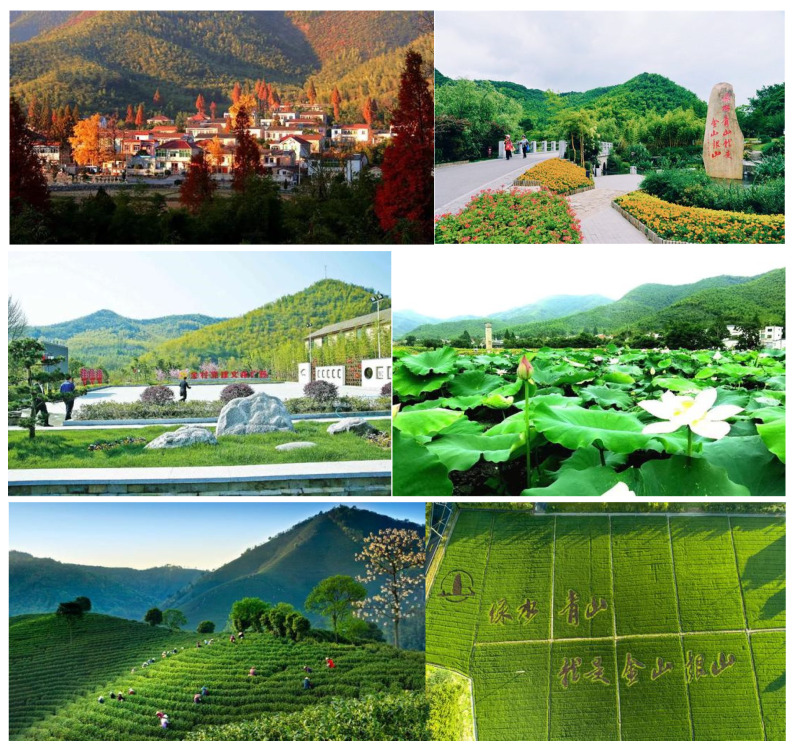
The current appearance of Yucun (the Chinese characters in the last picture are “green mountains are gold mountains”).

**Figure 6 ijerph-19-07580-f006:**
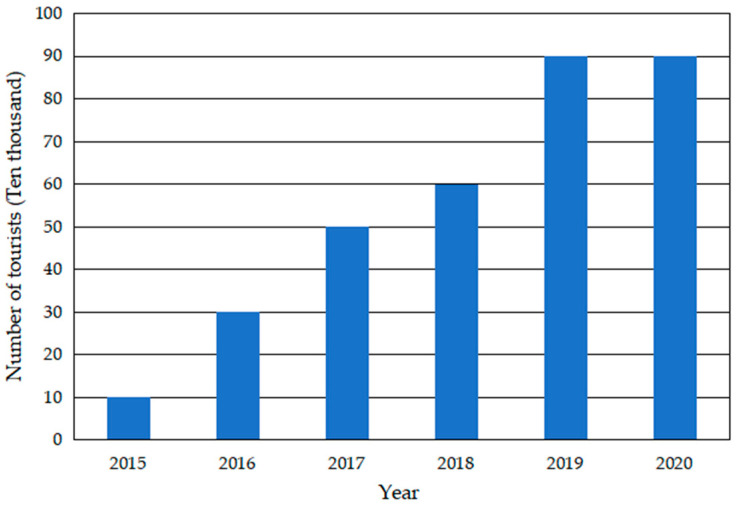
The number of tourists visiting Yucun from 2015 to 2020.

**Figure 7 ijerph-19-07580-f007:**
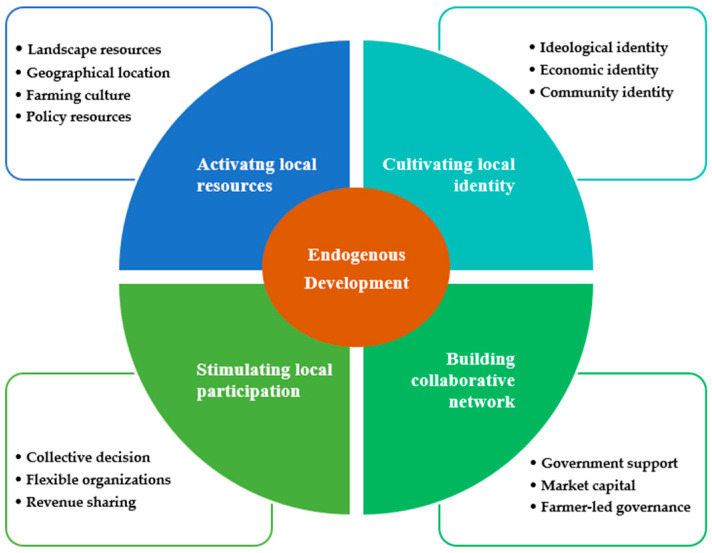
The endogenous development framework of Yucun’s green economy.

**Figure 8 ijerph-19-07580-f008:**
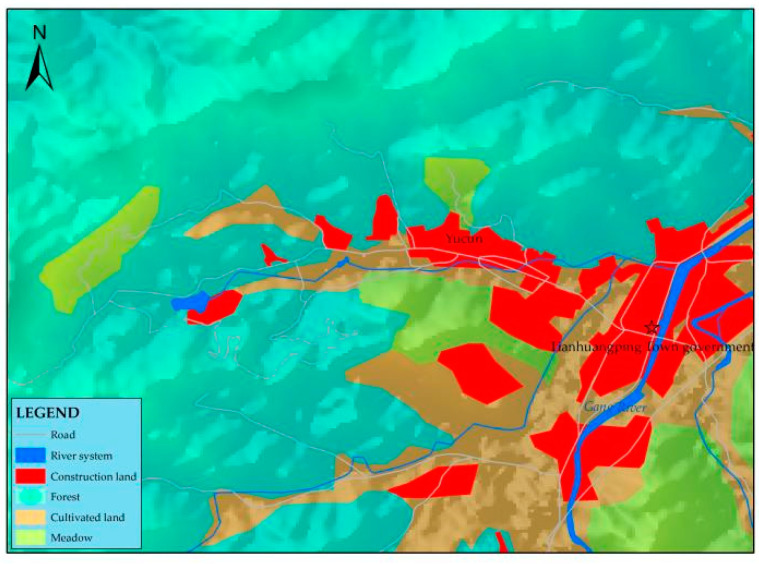
A schematic diagram of Yucun’s geographical resources.

**Table 1 ijerph-19-07580-t001:** Positive changes took place in Yucun.

Dimension	In the 1990s	Year of 2020
Industrial and employment structure	Mining and cement factories Only 80 people worked in the tertiary industry	Tourism, catering and other services More than 500 people work in the tertiary industry
Infrastructure and housing construction	❖ The main roads were only 4 m wide and pockmarked ❖ Lack of professional logistics services ❖ Small single-room bungalow	❖ The main roads are 12 m wide and the hardening rate reaches 100% ❖ E-commerce service stations and express delivery into the village ❖ A two- or three-story cottage
Income channels and social insurance	A single source of income No social insurance	Dividends, rent, salary, profit, etc. Endowment insurance, medical insurance, etc.
Lifestyle and environment	❖ Serious pollution and poor quality air ❖ Garbage recycling rate was below 50% ❖ Sewage disposal rate was below 60% ❖ Greening rate was below 10% ❖ Few people over 80 years old	❖ Pleasant living environment ❖ Garbage recycling rate is over 90% ❖ Sewage disposal rate is over 90% ❖ Greening rate is over 80% ❖ The number of people aged 80 and above increases by 3.2 times
Community governance	Many conflicts and disputes among villagers Imperfect governance system Low efficiency in dispute resolution	Establish multiple self-governing organizations Establish a normative system No criminal case, no mass petition, and no leapfrog petition

**Table 2 ijerph-19-07580-t002:** The results of open coding.

Concepts	Categories
Natural landscape (a1); Man-made landscape (a2); Agricultural and sideline products (a3);	Landscape resources (A1)
Traffic circle (a4); Road quality (a5); Traffic time (a6); Means of transportation (a7);	Geographical location (A2)
Cultural and creative products (a8); Farming experience (a9); Local customs (a10); Local culture (a11); Historical buildings (a12);	Farming culture (A3)
Project application (a13); Financial subsidy (a14);	Policy resources (A4)
Value judgement (a15); Development ideas (a16); Guidance of village cadres (a17);	Ideological identity (A5)
Income increase (a18); Income channels (a19); Confidence in future earnings (a20);	Economic identity (A6)
Village regulations (a21); Community attachment (a22); Common memory (a23);	Community identity (A7)
Democratic election (a24); Democratic decision-making (a25); Democratic management (a26); Democratic supervision (a27); Openness of affairs (a28);	Collective decision (A8)
Villagers’ congress (a29); Joy and bereavement events council (a30); Moral evaluation council (a31); Anti-drug and gambling council (a32); Peace volunteer team (a33);	Flexible organizations (A9)
Distribution to household (a34); Social security system (a35)	Revenue sharing (A10)
Financial fund (a36); Chief responsibility system (a37); Promotion mechanism (a38); Project-driven (a39);	Government support (A11)
Introduction of market players (a40); Business environment (a41); Mobilizing the enthusiasm of market players (a42); Benefit distribution mechanism (a43);	Market capital (A12)
Respecting the wishes of the villagers (a44); Improving the guidance mechanism for villagers’ participation (a45); Cooperation between government and villagers (a46).	Farmer-led governance (A13)

**Table 3 ijerph-19-07580-t003:** The results of axial coding.

Primary Categories	Second Categories
Activating local resources (AA1)	Landscape resources (A1); Geographical location (A2); Farming culture (A3); Policy resources (A4)
Cultivating local identity (AA2)	Ideological identity (A5); Economic identity (A6); Community identity (A7)
Stimulating local participation (AA3)	Collective decision (A8); Flexible organizations (A9); Revenue sharing (A10)
Building collaborative network (AA4)	Government support (A11); Market capital(A12); Farmer-led governance (A13)

**Table 4 ijerph-19-07580-t004:** The importance of endogenous development and the representative views of previous scholars on the effects of rural tourism.

Core Category	Dimension	Importance	Representative Views of Previous Scholars on the Positive Effects of Rural Tourism	Representative Views of Previous Scholars on the Negative Effects of Rural Tourism
Endogenous development	Activating local resources	Fully tapping the potential of local resources can create more opportunities. Sustainable development can only be ensured by the conservation of local resources.	Rural tourism reduces residents’ dependence on natural resources [31]. Rural tourism promotes the preservation of traditional agricultural landscapes and local knowledge [32].	Improper exploitation of tourism leads to environmental degradation [8]. The short-term benefits of tourism make the poor tend to give up traditional industries, leading to the simplification of income channels [10].
	Cultivating local identity	Identity is the base of collective action. Only by gaining the recognition of the residents can the collective obtain the maximum resultant force and the minimum resistance.	❖ Rural tourism brings vitality and confidence to remote villages [32]. ❖ Rural tourism improves community capacity [33]. ❖ Rural tourism strengthens local community pride [46].	❖ Rural tourism may cause unfair distribution and lead to the decline of residents’ willingness to protect the ecology [9]. ❖ Rural tourism may lead to the collapse of neighborhood relations and conflicts of interest [10].
	Stimulating local participation	Only by ensuring residents’ rights to know and rights to express themselves can the collective make correct decisions. Only by sharing dividends equally can sustainable development be achieved.	Residents’ participation in rural tourism increases opportunities for future employment and entrepreneurship [47]. Rural tourism promotes residents’ awareness and participation in environmental protection [31].	The poor can only get few benefits from tourism, but bear a lot of costs [13]. The seasonality of tourism results in fluctuating income [46]. Rural tourism may only benefit some residents, while others may choose to be neutral or opposed [48].
	Building collaborative network	The proper introduction of external subjects and resources can make up for the shortage of internal resources and reduce the pressure of internal subjects.	❖ Rural tourism increases government revenue and benefits the poor through redistribution [49]. ❖ Tourism promotes the connection between the countryside and the outside world [46].	❖ External forces often have absolute control over local resources [14]. ❖ Rural tourism may cause economic pressure on villagers because of high investment and risk [13].

## Data Availability

Not applicable.

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
