# Peer review of "Developing Village-Based Green Economy in an Endogenous Way: A Case Study from China"

_ijerph, 2022, doi:10.3390/ijerph19137580_

Round 1
Reviewer 1 Report
The problems discussed in the article, although generally not new, as the authors point out in their paper, introduce a new perspective to the scientific discourse of its analysis based on the theory of endogenous development. So far in the literature the issue of rural tourism development has raised many controversies. Researchers have proved its positive and negative influence on both the environment and local community. However, they did not indicate specific answers for practical solution of the controversies arising around the development of this function of tourism. Particularly valuable is the identification of four dimensions of endogenous development of green economy, and the development of a framework model of endogenous development of green economy based on rural tourism. This model can also be empirically verified in the future in regions with different levels of economic development.
The content of the reviewed paper is important both for the development of scientific discourse on exemplary models of development of "green" economies" not only in rural areas and not only based on tourism. The presented content proves that the known and seemingly recognized research problems can still be looked at from new perspectives, based on diverse theories, thanks to which they can be even better recognized.
The quality of the presented study is high. Both in terms of content and form. The choice of research method for the analysis of the research problem defined in the study, which was the discovery of the endogenous way of green economy development at the village level through the method of case study. The proper application of the research method allowed us to identify the key dimensions of transforming ecological assets into economic benefits, some in an extremely careful manner. Each stage of the conducted case study analysis of Yucun's development was discussed in great detail with an indication of the value it brought to the whole process of changing the economy of this place.
The authors, through their thorough approach to the research conducted, have demonstrated extraordinary diligence. The theoretical part of the study provides a basis for empirical considerations and is closed with well-grounded conclusions being a valuable contribution to the development of both theory and practice.
The reviewed study is very interesting material not only for scientists who can find in it hints for further research but also for regions that looking for their own ways of development or changes in the directions of development, can draw a lot of knowledge from it.
Undoubtedly, the article is a part of the discourse that has been going on for years about the controversies connected with the transformation of ecological values of regions, mainly rural ones, into economic benefits. The main contribution of the reviewed paper to the ongoing scientific discourse is to indicate the criteria on the basis of which the transformation leads to a balance of environmental, social and economic interests of regions.
Author Response
Thanks very much for the reviewer's positive note. We highly appreciate that the reviewer considered the topic is very interesting and the quality of the presented study is high. This is a great encouragement to us. We will make further efforts to carry out more in-depth research in this field in the future.
Reviewer 2 Report
The article is theoretical and empirical in nature. Both the problem, the subject of research and the method were chosen quite accurately. The text can be improved as follows:
1. On the theoretical layer, I would suggest paying attention to an important factor of endogenous development, which is the entrepreneurship of villagers. REST (recreation, entertainment, sport and tourism) is an entrepreneurial process where innovation and change are key elements in seizing opportunities, gaining a competitive advantage and developing rural tourism. REST can be an important factor in endogenous development, especially in rural regions with a low degree of urbanization, etc. Entrepreneurship can be included in the "Activation of local resources"
2. In the empirical layer, in the context of "Building a cooperation network", I would suggest paying attention to the concept of economic activation of rural tourist regions based on the development of green economy clusters and the development of small and medium-sized enterprises (SMEs).
Author Response
Thanks for the reviewer's positive comments. We highly appreciate that the reviewer considered the subject of research and the method were chosen quite accurately. We also thank the reviewer for the revision suggestions. We've added some content on entrepreneurship in REST (recreation, entertainment, sport and tourism) in the "Activate local resources" section. As the reviewer said, villagers' entrepreneurship is an important way to activate local resources, and also an important way to realize the endogenous development of rural tourism, which can be verified from the practice of Yucun. We have also added some statements on green economy clusters and the development of small and medium-sized enterprises in the “Building collaborative network” section.
Reviewer 3 Report
Manuscript, IJERPH-1764747, review
Journal: International Journal of Environmental Research and Public Health
Type of manuscript: Article
Title: Developing Village-based Green Economy in an Endogenous Way: A Case from China
COMMENTS AND SUGGESTIONS FOR AUTHORS
The research work argues a case study of endogenous development (the practice of green economic development at the rural level). The paper highlights the three basic elements of resources, participation and identification as fundamental elements.
The link between ecotourism and environmental protection is an actual problem all over the world.
The balance between environmental protection and economic development at the local village level is the right scale to deal with the problem.
The transformation of rural ecological advantages into economic benefits, activating local resources, cultivating local identity and stimulating local participation is the right way to shape sustainable communities.
The authors point out that the theory of endogenous development takes internal and external resources at the local level and argues that the decision-making process must be locally shaped to ensure local interests in rural development.
I think that regardless of the emphasis on an important aspect of endogenous development (that is not identical to sustainable development) the goal is very similar and the paper should mention more directly the sustainability of development. I suggest that the concept of sustainable development be included in keywords so the paper would be more visible and consequently more quoted.
The paper gives definitions of green development and endogenous development, but I think they should be moved to the initial part of the text
I think the description in lines 102 to 105 is not necessary
In global scientific communication, parts of the text such as lines 130-140 are not common. It is clear that politics has a role to play in shaping development, but this is too explicitly emphasized. In Eastern Europe (where I came from) this is how it should have been written (politically correct) fifty years ago. I understand if this is necessary for the author, but I suggest that it is possible to reshape such parts of the text to make them less emphasized (closer to the usual scientific exposition).
It seems to me that Figure 2 is not needed to explain the topic of the work. Figure 3 is enough. It would be better to show the overall demographic trends with separate elements: birth rate, mortality, immigration and emigration.
It would be great to find a recent photo of the same place as figure 4. to confront it with the old one.
Politicians are too many times named in the paper. Scientific work is not a chronicle of the communist party. Example, line 323 (I am sorry, but I had to say it in that way).
Table 1. Replace NOW with year
Figure 8. Does not have a resolution good enough
Other notes for improvement of the research
In future research efforts, it is recommended to consult the papers:
D'Amato, D., Droste, N., Allen, B., Kettunen, M., Lähtinen, K., Korhonen, J.,& Toppinen, A. (2017). Green, circular, bio economy: A comparative analysis of sustainability avenues. Journal of cleaner production, 168, 716-734.
Winter, M. (2003). Embeddedness, the new food economy and defensive localism. Journal of rural studies, 19(1), 23-32.
Scientific contribution
The research findings of this paper are in line with the theoretical framework, which strengthens the comprehensiveness of endogenous development theory.
I think the work is worth publishing (with mentioned modifications and additions).
Kind regards
Author Response
Thanks for the reviewer's affirmation and recognition of the topic and content of this article.
As suggested by the reviewers, we added "sustainable development" to the key words, so the paper would be more visible and consequently more quoted.
The initial part of the text mainly introduces the background of green development and the purpose of this paper is to reveal how to effectively develop green economy at the village level from the perspective of endogenous development theory. The second section is to establish the theoretical framework of this paper. We tend to put the definitions of green economy and endogenous development in this part, which helps to ensure the logical integrity and clarity of the Conceptual Background.
We have deleted the description in lines 102 to 105.
In China's context, politics is important for spreading the idea of green development. We are not deliberately exaggerating the role of politics, just stating the facts in China objectively. However, the reviewer's advice was good, and we made some changes to the expression of this paragraph. We have deleted lines 133-139 and added three sentences in this paragraph that closer to the usual scientific exposition.
In China, rural land is collectively owned. Village collective members jointly own arable land and construction land. The income generated by collectively owned land and collectively funded investment projects constitutes the village collective income. Part of the collective income of the village will be distributed as dividends to the villagers, and part will be used for the construction of public facilities. In addition to collective dividends, personal disposable income can also come from other sources. Therefore, we prefer to keep both Figure 2 and Figure 3. Besides, during our investigation, the cadres of Yucun said that they did not have accurate data records about birth rate, mortality, immigration and emigration, so they could not provide them to us. For the focus of the article, we are more concerned with economic changes. Therefore, the data of these indicators cannot be obtained, which will not have a constraint on our research.
The place shown in Figure 4 is included in the first photo of Figure 5. We are currently unable to provide a picture taken at the same angle and distance.
The village cadre mentioned in line 323 is the party branch secretary of Yucun. In China, the village party secretary has the power and responsibility to manage and make decisions on the day-to-day affairs of the village. Here we just want to state an objective fact. The village secretary at that time proposed that Yucun should close some mines and factories to explore green development. We did not deliberately promote anything for the Communist Party. We have always maintained a neutral position as scholars in the writing process.
In Table 1, we have replaced NOW with year of 2020.
For Figure 8, we replaced it with a higher resolution. The original image was 1,142 by 834 pixels and the new image was 2,788 by 2,124 pixels.
Thanks to the reviewer for providing two important references for our future research. We will study these articles carefully and strive to do better in future research.
Thanks for the reviewer’s recognition of the scientific contribution of our article and that our article is worthy of publication. Thanks again for the reviewer’s revision suggestions and support.